# Synthesis, Crystal Structure, DFT Studies, Docking Studies, and Fluorescent Properties of 2-(Adamantan-1-yl)-2*H*-isoindole-1-carbonitrile

**Jacques Joubert** 

Pharmaceutical Chemistry, School of Pharmacy, University of the Western Cape, Private Bag X17, Bellville 7535, South Africa; jjoubert@uwc.ac.za; Tel.: +27-21-959-2195

**Abstract:** 2-(Adamantan-1-yl)-2*H*-isoindole-1-carbonitrile (**1**) has been identified as a neurobiological fluorescent ligand that may be used to develop receptor and enzyme binding affinity assays. Compound **1** was synthesized using an optimized microwave irradiation reaction, and crystallized from ethanol. Crystallization occurred in the orthorhombic space group $P2_12_12_1$ with unit cell parameters: a = 6.4487(12) Å, b = 13.648(3) Å, c = 16.571(3) Å, V = 1458(5) Å$^3$, Z = 4. Density functional theory (DFT) (B3LYP/6-311++G (d,p)) calculations of **1** were carried out. Results indicated that the optimized geometry was similar to the experimental results, with a root-mean-squared deviation of 0.143 Å. In this paper, frontier molecular orbital energies and net atomic charges are discussed with a focus on potential biological interactions. Docking experiments within the active site of the neuronal nitric oxide synthase (nNOS) protein crystal structure were carried out and analyzed. Important binding interactions between the DFT-optimized structure and amino acids within the nNOS active site were identified that explained the strong NOS binding affinity reported. Fluorescent properties of **1** were studied using aprotic solvents of different polarities. Compound **1** showed the highest fluorescence intensity in polar solvents, with excitation and emission maximum values of 336 nm and 380 nm, respectively.

**Keywords:** 2-(Adamantan-1-yl)-2H-isoindole-1-carbonitrile; X-ray diffraction; density functional theory (DFT); molecular orbital calculations; fluorescence; docking; neuronal nitric oxide synthase (nNOS); fluorescent ligand

## 1. Introduction

Radioligand binding techniques are extensively used to explore biological proteins involved in the pathophysiology of neurodegenerative disorders, such as Alzheimer's and Parkinson's disease [1–3]. Neuroprotective targets, including the neuronal nitric oxide synthase (nNOS) enzyme [4], the *N*-methyl-D-aspartate (NMDA) receptor [3], and voltage gated calcium channels (VGCC) [5] have been widely studied using radioligands. Radioligand binding techniques have also afforded a high level of usefulness and sensitivity in the study of pharmacological proteins. However, the use of alternative approaches, such as fluorescent methods, to study enzyme-ligand and receptor-ligand binding interactions can provide information that may not be readily available using conventional radiopharmacology techniques, and can also avoid some of the disadvantages associated with radiopharmacology, such as high costs, health hazards, disposal, and possible methodological implications [6–9]. The development of small molecule fluorescent probes to study neurobiological events has thus become a research focus area that has received considerable attention [9,10]. Currently, there are no commercially available fluorescent ligands that are able to directly bind to the nNOS enzyme, NMDA receptor, and/or VGCC that can be used to develop binding affinity studies using fluorescent displacement assays.

2-(Adamantan-1-yl)-2*H*-isoindole-1-carbonitrile (**1**, Scheme 1) has shown the ability to bind to and inhibit the nitric oxide synthase (NOS) enzyme [11] and antagonize the NMDA receptor [12] and block VGCC [12]. Compound **1** also has inherent fluorescent properties because of the cyanoisoindole fluorophore in its structure [11,13]. The ability of **1** to bind to these targets and show strong fluorescence may enable the design of a neurobiological fluorescent ligand that could be used to develop receptor and enzyme binding affinity assays. This study therefore set out to improve the synthesis of **1**, study its crystal structure and geometry in both the solid and free-form state, analyze the frontier molecular orbitals and atomic net charges, perform docking experiments to elucidate the potential biological binding interactions, and further explore its fluorescent properties. These studies will provide valuable information that may be used to develop fluorescent displacement assays utilizing compound **1**.

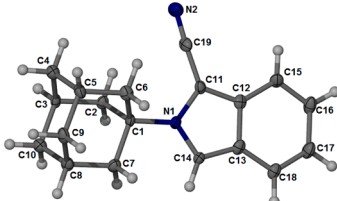

**Scheme 1.** Reagents and conditions for the synthesis of 2-(adamantan-1-yl)-2H-isoindole-1-carbonitrile (**1**): (i) MeOH, NaCN, $H_2O$, MW, 150 W, 100 °C, 150 psi, 10 min, 91% yield. OPD: o-phthaldialdehyde.

## 2. Results and Discussion

### 2.1. Chemistry

2-(Adamantan-1-yl)-2H-isoindole-1-carbonitrile (**1**) was obtained through the conjugation of amantadine HCl with *o*-phthaldialdehyde (OPD) in the presence of sodium cyanide, under optimized microwave (MW) irradiation (150 W, 100 °C, 150 psi) conditions (Scheme 1). Both OPD and amantadine HCl are commercially available. The reaction proceeded efficiently, providing the title compound in an excellent yield of 91% after only 10 min of reaction time. This is a significant improvement in yield and reduction in reaction time compared to the conventional method previously described, where a yield of 66% was obtained with a reaction time of 24 h [12]. The molecular structure of **1** was confirmed by $^1$H and $^{13}$C nuclear magnetic resonance (NMR), high-resolution mass spectroscopy, and X-ray diffraction.

### 2.2. Crystal Structure and Geometry Optimization

A crystalline material was grown from EtOH at room temperature for 24 h. The rod-like crystal crystallized in the orthorhombic space group, $P2_12_12_1$. The asymmetric unit cell of compound **1** contained four molecules. The structure of **1** is shown in Figure 1, and the molecular crystal packing image is shown in Figure 2a,b. Selected bond lengths and bond angles are listed in Table 1.

**Figure 1.** The asymmetric unit of compound **1** showing the atomic numbering scheme and 30% probability displacement ellipsoids of non-H atoms.

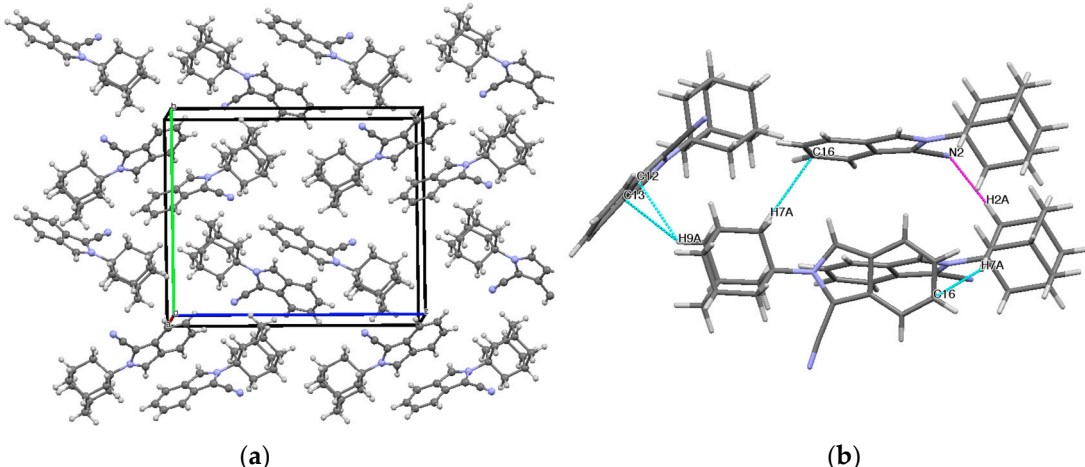

|  | (a) |  | (b) |

**Figure 2.** (**a**) The molecular crystal packing of compound **1** viewed along the b-axis. (**b**) Partial crystal packing showing the H-bond (magenta) and C-H···π (light blue) intermolecular interactions along the c-axis.

**Table 1.** Bond lengths (Å), angles (°), and theoretical calculations of compound **1**.

| Bond Lengths (Å) | X-ray Crystal | DFT [a] | Bond Angles (°) | X-ray Crystal | DFT [a] |
|---|---|---|---|---|---|
| N1-C14 | 1.352 | 1.360 | C14-N1-C11 | 108.77 | 109.13 |
| N1-C1 | 1.498 | 1.499 | C1-N1-C14 | 126.57 | 125.31 |
| N1-C11 | 1.389 | 1.395 | C2-C3-C10 | 110.53 | 109.52 |
| N2-C19 | 1.136 | 1.161 | C10-C3-C4 | 110.10 | 109.74 |
| C3-C10 | 1.520 | 1.539 | N1-C1-C2 | 109.54 | 109.72 |
| C1-C2 | 1.531 | 1.548 | C2-C1-C6 | 110.85 | 110.21 |
| C11-C12 | 1.397 | 1.408 | C6-C1-C7 | 108.39 | 108.39 |
| C12-C13 | 1.433 | 1.433 | C11-C12-C13 | 106.98 | 106.78 |
| C13-C18 | 1.412 | 1.417 | C11-C12-C15 | 132.27 | 132.92 |
| C5-C6 | 1.526 | 1.542 | C13-C12-C15 | 120.67 | 120.30 |
| C15-C16 | 1.375 | 1.375 | N1-C11-C12 | 107.88 | 107.88 |
| C16-C17 | 1.408 | 1.425 | N1-C11-C19 | 125.49 | 125.69 |
|  |  |  | C13-C18-C17 | 118.66 | 118.44 |
|  |  |  | N2-C19-C11 | 176.20 | 176.79 |
|  |  |  | C12-C15-C16 | 117.31 | 118.21 |
|  |  |  | C15-C16-C17 | 122.20 | 121.68 |

[a] Density functional theory (DFT) optimization was performed at B3LYP/6-311++G(d,p) level.

Bond lengths, angles, and torsion angles of the adamantane ring system were consistent with other previously reported compounds [14]. The geometry of the cyano moiety was within the usual range for nitriles. Generally, the bond lengths and bond angles of the isoindole structure were comparable to those reported for N-methylisoindole [15], suggesting that substitutions to the pyrrolidine ring at the C11 position do not significantly alter the solid-state geometry. The orientation of the cyanoisoindole moiety in relation to the adamantane moiety is defined by the bond angles of C1-N1-C11 (124.63°) and C1-N1-C14 (126.56°), as well as the torsion angles of C1-N1-C11-C12 (-177.56°) and C1-N1-C14-C13 (178.93°). The crystal packing is not planar and has an inversion-related molecular arrangement. An intermolecular hydrogen bond is observed at C2-H2A···N2. The D···A distance is 2.61 Å and the D-H···A angle is 161°. The formation of the self-assembled crystal structure can also be attributed to a number of weak C-H···π interactions between the adamantane and the isoindole moieties of the molecules constituting the crystal assembly (Figure 2b). Thus, the crystal structure is built up through a combination of hydrogen bond and C-H···π interactions.

A density functional theory (DFT) geometry optimization of **1** was performed with the Gaussian09 (Gaussian Inc., Wallingford, UK, 2016) program package [16] using Becke's three parameters Lee–Yang–

Parr exchange correlation functional (B3LYP). This calculation combines the hybrid exchange functional of Becke [17] with the gradient-correlation functional of Lee, Yang, and Parr [18] using the 6-311G++(d,p) basis set. Previous studies have shown that gas phase calculations correlate well with crystal structures; therefore, no solvent corrections were made [19]. The X-ray refined data were used to select a crystal unit containing the starting geometries of compound **1**. The geometry of compound **1** was optimized, and the DFT-optimized molecule was generated. Table 1 contains the data comparing the experimental and calculated structure of **1**.

The C-C bond lengths within the isoindole structure was in the range of 1.41–1.42 Å and 1.42–1.43 Å for the crystal structure and at the calculated B3LYP/6-311G++(d,p) level, respectively. This is considerably shorter than typical C-C single bonds (1.54 Å) [20]. The experimental (1.36–1.40 Å) and calculated (1.37–1.41 Å) C-C bonds of the adamantane structure were within the reported characteristic range [14]. The C=C bonds were found to be marginally longer than standard C=C bond lengths (1.34 Å) [20] for both the experimental (1.52–1.54 Å) and calculated (1.53–1.55 Å) results. For N2-C19, the calculated C≡N (nitrile) bond length was 1.16 Å, which is longer than the experimental value of 1.14 Å and is exactly the same as the bond length (1.16 Å) described for nitriles in chemistry literature. The N1–C1, N1–C11, and N1–C14 bond lengths were calculated as 1.50, 1.40, and 1.36 Å at the B3LYP/6-311G level, and the crystal structure values were 1.49, 1.39, and 1.35 Å, respectively. Normal C-N bond lengths were reported as 1.47 Å; however, the N1–C11 and N1–C14 bond lengths of the isoindole structure were considerably shorter, while N1–C1 was nearer to the normal length (Table 1). The shorter bond lengths occurred because they were part of the unconjugated isoindole structure. These bond lengths were consistent with similar isoindole structures reported in the literature [15]. The calculated bond angles correlated well with that of the crystal structure with a maximum deviation of 1.26° for C1–N1–C14, which is calculated as 125.31°, and the crystal structure value was 126.57°. The largest deviation between bond lengths was 0.025 Å for N2–C19.

These deviations between bond lengths and bond angles, although relatively minor, occurred because the experimental results represented the molecule in its solid state and the calculations represented the molecule in its gaseous state. These differences in bond parameters can be ascribed to the intermolecular interactions and crystal field that has linked the molecules together in its solid state, thus changing the physical and chemical parameters, compared to the molecule in its gaseous state. Overlay analysis was performed using YASARA version 18.11.21 (YASARA Biosciences GmbH, Vienna, Austria, 2018) to globally compare the structure obtained with the theoretical calculation with the experimental crystal structure (Figure 3). A relatively low root-mean-squared deviation (RMSD) value of 0.143 Å was obtained, confirming that there were minor differences between the calculated and experimental structures.

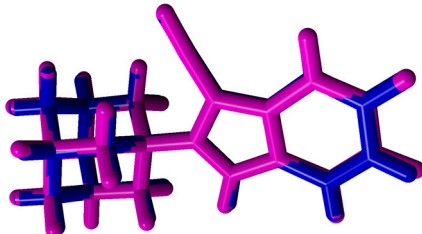

**Figure 3.** Atom-by-atom superimposition of DFT-optimized compound **1** (magenta) calculated using B3LYP/6-311++G (d,p) on the X-ray structure (blue) of **1** (RMSD = 0.143 Å). DFT: density functional theory.

### 2.3. Frontier Molecular Orbitals and Atomic Net Charges

The Highest Occupied Molecular Orbital (HOMO) and the Lowest Unoccupied Molecular Orbital (LUMO) are the most significant orbitals found in molecules. HOMO has electron-donating ability, and LUMO is able to obtain an electron [21]. According to the frontier molecular orbital theory, HOMO and LUMO are significant factors that affect the biological activity, molecular reactivity, ionization,

and electron affinity of molecules [22–27]. Thus, frontier orbital energy studies can provide valuable insight into the potential biological mechanisms of biologically active compounds. In Figure 4, the distribution and energy levels of HOMO and LUMO orbitals computed at the B3LYP/6-311G level for compound **1** are shown.

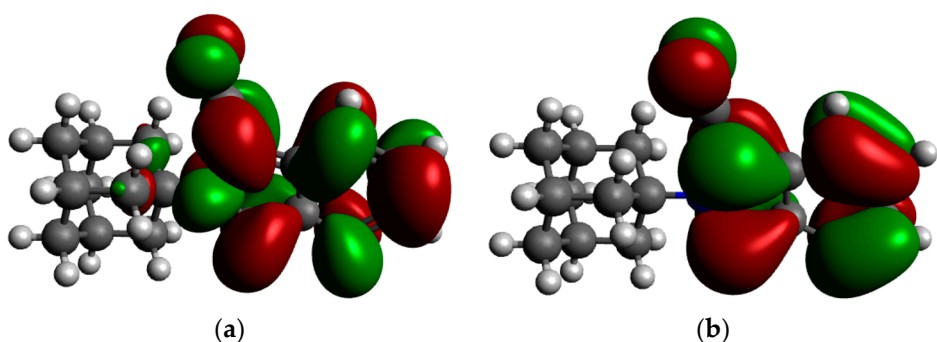

(a)          (b)

**Figure 4.** The frontier molecular orbitals of compound **1** calculated using B3LYP/6-311++G (d,p). (**a**) LUMO (−1.550 eV), (**b**) HOMO (−5.625 eV).

As observed in Figure 4, both the LUMO and HOMO is predominantly localized at the nitrile and isoindole moieties. This indicates that the activity associated with this molecule could generally be attributed to the cyanoisoindole structure, with the adamantane moiety mostly providing structural bulk and/or lipophilic function. This finding is in line with hypotheses from previous studies [11,12]. It is thus postulated, based on HOMO and LUMO findings, that the cyanoisoindole structure will form interactions with amino acids lining the active sites of the NOS enzyme, NMDA receptor, and/or VGCC, and in so doing, give rise to its reported biological activity.

The atomic net charges of the DFT-optimized structure are shown in Figure 5. The two N-atoms have the highest electronegativity and showed the lowest net atomic charge values (N1 = −0.32 e and N2 = −0.19 e) within the structure of **1**. Therefore, it is expected that the N-atoms will be the most favored to interact with positively charged amino acids within the enzyme and receptor active sites. However, even though the N1 atom showed a lower net atomic charge compared to N2, it will encounter some difficulty to form interactions with amino acids because of the structural hindrance imposed by the bulky adamantane group. Therefore, interactions are expected to be present at the N2 position because the nitrile group is protruding outwards from the isoindole structure, and should present with an effective handle for interactions to take place.

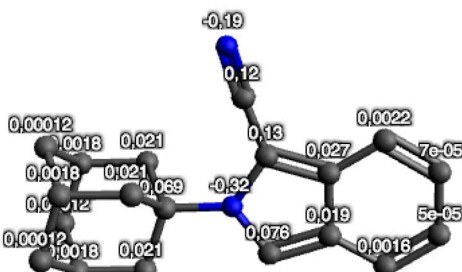

**Figure 5.** The atomic net charges e of the DFT-optimized structure of **1**. Hydrogens are not shown.

## 2.4. Docking Studies

Molecular modelling studies using the NMDA receptor and VGCC were not carried out because the full protein crystal structure of these targets have not yet been published. Therefore, in order to further explore the interaction profile and biological potential/application of **1**, molecular docking experiments were performed using the crystal structure (Protein Data Bank I.D.: 1OM5) of rat nNOS

co-crystallised with the known nNOS inhibitor, 3-bromo-7-nitroindazole (3B7NI) [28] (Figures 6 and 7). 3B7NI crystallized in the active heme (HEM750) site domain of nNOS.

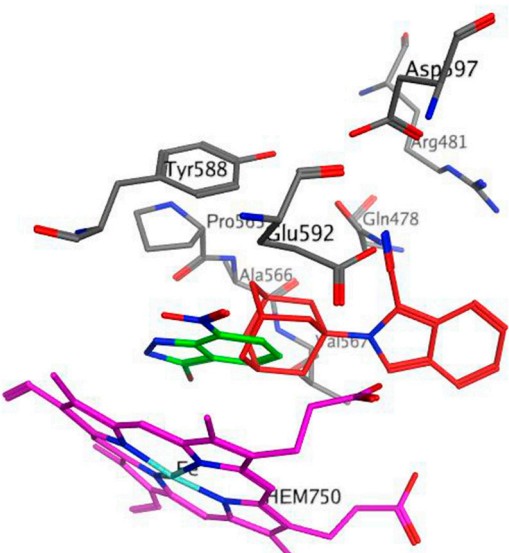

**Figure 6.** The putative binding modes and orientation of 3-bromo-7-nitroindazole (3B7NI) (**green**) and compound **1** (**red**) within the rat neuronal nitric oxide synthase (nNOS) heme domain active site. The heme cofactor (HEM750) is shown in magenta, and the surrounding amino acids are shown in grey.

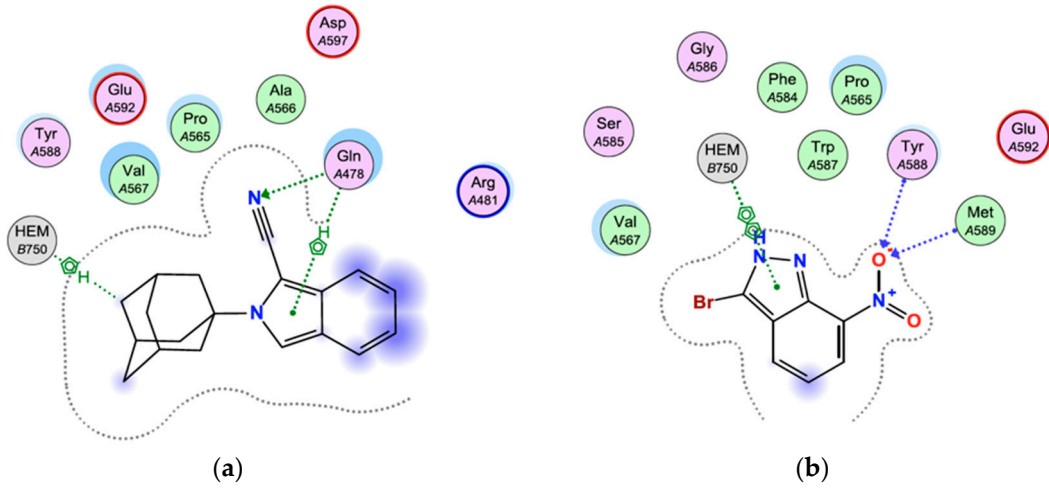

**Figure 7.** Binding interactions and binding affinity (kcal/mol) of compound **1** and the co-crystallized ligand, 3B7NI, within the active site of the nNOS heme domain. (**a**) Binding affinity (**1**) = −5.722 kcal/mol, (**b**) Binding affinity (3B7NI) = −6.000 kcal/mol.

Inspection of the binding mode of 3B7NI indicated that two hydrogen bond contacts were formed through the interaction of the 7-NO$_2$ moiety with residues Tyr588 and Met589 (Figure 7). A π-π interaction was also observed between the pyrazole ring of the indazole structure and HEM750. These interactions within the active site could be responsible for the potent nNOS inhibitory activity reported for 3B7NI (rat nNOS IC$_{50}$ = 0.17 μM) [28]. The binding affinity of 3B7NI was found to be −6.000 kcal/mol. The DFT-optimized (B3LYP/6-311++G (d,p)) structure of **1** was used for further docking studies (Figure 7). Compound **1** was docked using the protocol as described in the experimental section and was able to access and bind to the active heme site domain of nNOS, similar to 3B7NI. Two H-π interactions were observed between the adamantane moiety and HEM750, and between Gln478 and the pyrrolidine ring of the isoindole structure. Compound **1** also showed a hydrogen bond interaction between the electronegative nitrile moiety and Gln478. This result was as

expected, based on the frontier molecular orbital and net atomic charges findings. The binding affinity of **1** was found to be $-5.722$ kcal/mol. The binding interactions and binding affinity of **1**, which is similar to 3B7NI, may be the reason for the significant NOS inhibitory activity reported (rat NOS $IC_{50}$ = 0.29 µM) [12]. A virtual derivative of **1** devoid of the nitrile group was also docked, and showed no binding interactions within the active site. This finding demonstrates the importance of the nitrile moiety in this structure for nNOS inhibitory activity. In addition, no covalent intermolecular biological interactions were identified, suggesting that this molecule will be suited for fluorescent displacement studies to determine the binding affinity of test compounds.

It is important to note that **1** has only been evaluated on a crude NOS extract [12] containing the inducible-, endothelial-, and neuronal NOS isoforms, and further biological studies are necessary in order to determine the NOS isoform selectivity profile of this compound. However, this docking study indicates that **1** should show significant nNOS inhibitory activity which is vital in order to treat neurodegenerative disorders or to develop molecular probes for neurobiological studies.

*2.5. Fluorescent Properties*

The excitation and emission spectra of compound **1** recorded in a dimethyl sulfoxide (DMSO) solution ($1 \times 10^{-5}$ mol/L) is shown in Figure 8a. A BioTek Synergy (BioTek Instruments, Inc., Winooski, VT, USA, 2013) fluorescent plate reader was used for all fluorescence measurements. The excitation absorption maximum and the emission maximum were found at 336 nm and 380 nm, respectively. The strong emission can be explained by comparing the fluorescent properties of the 1-nitrile substituted compound **1** with that of a reported unsubstituted N-methylisoindole compound (emission = 345 nm) [15]. When the electron-withdrawing nitrile group is present, a red-shift in the emission maximum and increase in fluorescence intensity is observed. This is because the charge separation of the cyanoisoindole moiety is increased due to an internal Stark effect [29–31]. Therefore, the nitrile moiety is of significant importance in order for compound **1** to show strong fluorescent properties.

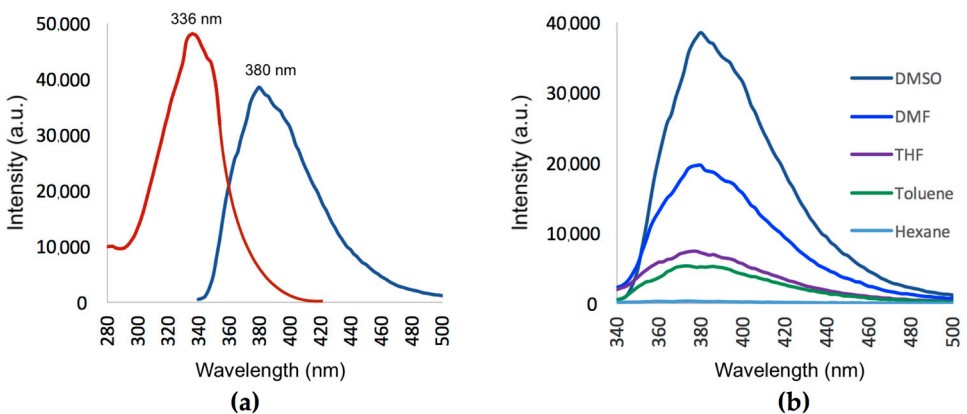

**Figure 8.** (**a**) Excitation and emission spectra of compound **1**. (**b**) Emission spectra of compound **1** showing the effect of different aprotic solvents of varying polarities.

Figure 8b shows the influence of different aprotic solvents of varying polarities on the emission spectra of compound **1** ($1 \times 10^{-5}$ mol/L). No noticeable shift in the emission maximum of **1** was observed in the different solvents. The results, however, indicate that the solvent polarity had a profound effect on the emission intensity of **1**. The fluorescence intensity significantly decreased as the polarity of the solvents decreased (Figure 8b). This initial study indicates that **1** exhibits a solvatochromic effect in the different solvents with varying polarity, most probably due to an internal charge transfer phenomenon.

Fluorescent properties of the solid crystalline state of **1** were also determined. Interestingly, a definite red shift was observed for both the excitation (336 to 346 nm) and emission (380 to 400 nm)

for the solid state of **1**. In the solid state, there were intermolecular interactions present between the molecules in the crystal field (see Figure 2b). These interactions altered the physical and chemical properties, which led to a change in the photochemical properties of **1** compared to when the compound was dissolved in organic solvents and intermolecular bonds were broken.

## 3. Materials and Methods

### 3.1. Chemistry

#### 3.1.1. General Information

Unless otherwise specified, all chemicals were obtained from Sigma-Aldrich or Merck. All the chemicals were of analytical grade and used without further purification. Compound **1** was characterized using nuclear magnetic resonance (NMR) and high-resolution mass spectrometry (HRMS) techniques. NMRs were obtained using a Bruker 400 MHz instrument. The chemical shifts are reported as δ values in ppm downfield from tetramethylsilane as an internal standard. Deuterated chloroform ($CDCl_3$) was used as NMR solvent. The multiplicities of NMR signals are expressed as: s, singlet or m, multiplet. HRMS was recorded on a Waters API Q-Tof Ultima (Waters Corporation, Milford, MA, USA, 2011) electro-spray ionization mass spectrometer at 70 eV and 100 °C. The melting point was determined using a Stuart SMP-10 (Cole-Palmer, Staffordshire, UK, 2010) melting-point apparatus. The melting point is uncorrected. Microwave synthesis was performed using a CEM Discover™ (CEM Corporation, Matthews, NC, USA, 2014) focused closed vessel (maximum capacity = 30 ml) microwave synthesis system.

#### 3.1.2. Synthesis of 2-(adamantan-1-yl)-2H-isoindole-1-carbonitrile (**1**)

Amantadine hydrochloride (0.600 g, 3.301 mmol) and sodium cyanide (NaCN, 0.132 g, 3.301 mmol) was dissolved in 15 ml methanol. Distilled water (1 mL) and *o*-phthaldialdehyde (0.442 g, 3.301 mmol) were added. The reaction mixture was placed in the microwave reactor and irradiated at 150 W, 100 °C, and 150 psi for 10 min. Upon cooling in an ice bath, a white precipitate formed and was filtered and washed twice with cold methanol (10 ml) and twice with distilled water (10 ml) to yield the pure product (1) as a white amorphous solid in high yield (0.828 g, 2.997 mmol, 91%). Further crystallization by slow diffusion of a solution in EtOH was carried out to provide a rod-like single crystal suitable for X-ray diffraction analysis. The structure elucidation data is similar to that previously reported [12]. Mp: 160 °C; [1]H NMR (400 MHz, $CDCl_3$) $\delta_H$: 7.69–7.64 (m, 2H), 7.51 (s, 1H), 7.26–7.19 (m, 1H), 7.12–7.04 (m, 1H), 2.45 (m, 6H), 2.32 (s, 3H), 1.85-1.80 (m, 6H); [13]C NMR (100 MHz, $CDCl_3$) $\delta_C$: 150.6, 133.9, 125.3, 122.7, 122.4, 120.9, 117.8, 116.6, 115.9, 60.4, 43.00, 35.9, 30.0; HRMS m/z: Calculated for $C_{19}H_{20}N_2$ (M + H); 276.16366, found 276.16265.

### 3.2. X-Ray Crystallography

Single-crystal X-ray intensity data were collected on a Bruker 3-circle Apex II DUO X-ray diffractometer equipped with an INCOATEC IµS HB microfocus sealed tube (MoKα radiation λ = 0.71073 Å) fitted with a multilayer monochromator. Data were captured with a charge-coupled device (CCD) area detector. Data collection was carried out at 100 K using an Oxford Cryosystems cryostat (700 series Cryostream Plus) attached to the diffractometer. Data collection and reduction were carried out using the Bruker software package, APEX3 [32], using standard procedures. All structures were solved and refined using SHELX-2016 [33] employed within the X-Seed [34,35] environment. Hydrogen atoms were placed in calculated positions using riding models. Table 2 contains the crystal data and refinement parameters.

**Table 2.** Crystal data and refinement parameters of the tile compound **1**.

| Crystal Data | 1 |
|---|---|
| Chemical formula | $C_{19}H_{20}N_2$ |
| $M_r$ | 276.37 |
| Crystal system, space group | Orthorhombic, $P2_12_12_1$ |
| Temperature (K) | 100 |
| $a$, $b$, $c$ (Å) | 6.4487 (12), 13.648 (3), 16.571 (3) |
| $V$ (Å$^3$) | 1458.4 (5) |
| $Z$ | 4 |
| Radiation type | Mo $K\alpha$ |
| $\mu$ (mm$^{-1}$) | 0.07 |
| Crystal size (mm) | $0.59 \times 0.05 \times 0.05$ |
| Diffractometer | Bruker *APEX* II DUO CCD |
| $T_{min}$, $T_{max}$ | 0.931, 1.000 |
| No. of measured, independent, and observed [$I > 2\sigma(I)$] reflections | 14040, 3372, 2528 |
| $R_{int}$ | 0.072 |
| $(\sin\theta/\lambda)_{max}$ (Å$^{-1}$) | 0.650 |
| $R[F^2 > 2\sigma(F^2)]$, $wR(F^2)$, $S$ | 0.074, 0.199, 1.04 |
| No. of reflections | 3372 |
| No. of parameters | 190 |
| $\Delta\rho_{max}$, $\Delta\rho_{min}$ (e Å$^{-3}$) | 0.69, −0.28 |

A CIF file containing complete information of the studied structure was deposited with CCDC, deposition number 1881043, and is freely available upon request from the Director, CCDC, 12 Union Road, Cambridge CB2 1EZ, UK (Fax: +44-1223-336033; e-mail: deposit@ccdc.cam.ac.uk) or from the following website: www.ccdc.cam.ac.uk/data_request/cif.

### 3.3. Theoretical Calculations

A crystal unit within the crystal structure of **1** was selected as the starting structure for the DFT calculations. DFT-B3LYP/6-311G++(d,p) methods in Gaussian09 [16] was used for structural optimization. Solvent corrections were not made for these calculations. Vibration analysis showed no negative eigenvalues, indicating that the optimized structure represents a minimum on the potential energy surface. For the optimized structure, the HOMO and LUMO were drawn using Avogadro 1.2.0 [36,37]. Atomic net charges were also calculated using Avogadro 1.2.0.

### 3.4. Docking Studies

The docking studies were performed using the rat nNOS heme domain crystal structure (Protein Data Bank I.D.: 1OM5). The docking method employed is similar to that reported by our group for studies on a protease enzyme [38] The Molecular Operating Environment (MOE) 2018 software suite [39] was used for docking studies with the following protocol. (1) The enzyme protein structure was checked for missing atoms, bonds, and contacts. (2) Removal of water molecules, 3D protonation, and energy minimization was carried out with parameters, force field: MMFF94X+solvation, gradient: 0.05, chiral constraint and current geometry. This minimized structure was used as the enzyme for docking analysis. (3) The DFT B3LYP/6-311++G (d,p) optimized structure of **1** was saved as a pdb file and imported into the MOE database. (4) Compound **1** was subsequently docked within the nNOS heme domain active site using the MOE Dock application. The active site was selected based on the proximity of the co-crystallized ligand, 3B7NI, with the help of the MOE Site Finder tool. The docking algorithm, which was chosen for these experiments, was based on induced fit docking to allow for flexible interactions of the test ligand with the protein. (5) The best binding pose of compound **1** was visually inspected, and the interactions with the binding pocket residues were analyzed. The selected parameters that were used to calculate the score and interaction of the ligand molecule with the nNOS enzyme were as follows: Rescoring function, London dG; Placement, Triangle matcher; Retain, 30;

Refinement, Force field; Rescoring 2, London dG. The build-in scoring function of MOE, S-score, was used to predict the binding affinity (kcal/mol) of the ligand with the enzyme protein active site after docking.

## 4. Conclusions

2-(Adamantan-1-yl)-2H-isoindole-1-carbonitrile (**1**) was synthesized using an optimized microwave irradiation synthesis method in high yield. The three-dimensional structure of **1** was confirmed by single crystal X-ray diffraction. Using DFT calculations at the B3LYP level of theory and the 6-311G++(d,p) basis set, the molecular structure and electronic properties of **1** were deduced. Molecular docking studies indicated important interactions of **1** with the active site of the nNOS enzyme and showed that the nitrile moiety is imperative for nNOS inhibitory activity. The fluorescent properties of **1** were studied, and strong fluorescence was shown at 380 nm in polar aprotic solvents.

This study has therefore provided valuable information of **1** that may be used to develop a pharmacological tool to investigate enzyme–ligand and/or receptor–ligand interactions utilizing modern fluorescent imaging techniques.

**Funding:** This research was funded by the National Research Foundation (NRF, South Africa, Grant Number: 11181) and the University of the Western Cape.

**Acknowledgments:** E. Antunes from the Chemistry Department of the University of the Western Cape is acknowledged for NMR data collection. L. Loots from the Chemistry and Polymer Science Department of Stellenbosch University is acknowledged for the X-ray diffraction data collection and analysis. The Centre for High Performance Computing at the Council for Scientific and Industrial Research (South Africa) is acknowledged for granting access to Gaussian09 software.

**Conflicts of Interest:** The authors declare no conflict of interest.

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
