# Peer review of "Synthesis, Crystal Structure, DFT Studies, Docking Studies, and Fluorescent Properties of 2-(Adamantan-1-yl)-2H-isoindole-1-carbonitrile"

_crystals, doi:10.3390/cryst9010024_

Reviewer 1 Report

The Authors present a study of the title compound, 2-(Adamantan-1-yl)-2H-isoindole-1-carbonitrile. They report an improved synthetic method, the crystal structure, fluorescent properties and computational studies. They also have investigated the interaction of the compound with the rat nNOS heme domain by means of computational docking experiments.  

The work is well conducted and it is of interest for the community working in the field. The information presented are useful and can lead to developments in the field of using pharmacological tools in enzyme/ligand interactions using fluorescent techniques.

I would just recommend a minor addition to improve the quality of the paper: from the DFT studies it seems that both homo and lumo are strongly localized on the isoindole moiety. What is the the effect of the adamantane moiety on the electronic properties? I suggest to run similar calculations for the isolated isoindole molecule to be compared with those of the compound discussed by the Authors in order to establish a possible effect of the adamantane on the electronic properties.

Author Response

Response 1

Reviewers comment: I would just recommend a minor addition to improve the quality of the paper: from the DFT studies it seems that both homo and lumo are strongly localized on the isoindole moiety. What is the the effect of the adamantane moiety on the electronic properties? I suggest to run similar calculations for the isolated isoindole molecule to be compared with those of the compound discussed by the Authors in order to establish a possible effect of the adamantane on the electronic properties.

Reply from author: We repeated the experiment using a cyanoisoindole structure (2H-isoindole-1-carbonitrile) that is not conjugated to the adamantane moiety. The results indicate that the homo and lomo is still strongly localised on the entire isoindole moiety. The homo and lomo distribution also did not change when compared to the adamantane substituted cyanoisoindole (compound 1 in the paper). Therefore, as mentioned in the manuscript, the adamantane moiety mostly provides structural bulk and/or lipophilic function and does not have a strong effect on the electronic properties of the molecule.

Reviewer 2 Report

The nNOS, NMDA and VGCC abbreviations appeared in text at Page 1 Line 40 before they have been deciphered at Page 1 Line 42.

 The calculated bond lengths and angles are in good agreement with experimental one’s despite of that the calculations related to the gas phase. It will be interesting to compare presented results with results of simulations using some code performing ab initio quantum mechanical calculations of solids.

 I can’t figure out why details of Materials and Methods placed after Results and Discussion section. Maybe they should be swapped?

Author Response

Response 1

Comment from Reviewer: The calculated bond lengths and angles are in good agreement with experimental one’s despite of that the calculations related to the gas phase. It will be interesting to compare presented results with results of simulations using some code performing ab initio quantum mechanical calculations of solids.

Reply from author: The above simulation suggested by the reviewer will not add significant value to the paper as the calculations (in gas phase) was found to be in good agreement with the experimental results. The use of gas phase calculations have been reported to correspond well with crystal structures. This has been mentioned in the paper and referenced accordingly (Section 2.2, and see reference 19). This type of calculation has also been found appropriate by other research groups. See: Sharma, A.; Jad, Y.E.; Ghabbour, H.A.; de la Torre, B.G.; Kruger, H.G.; Albericio, F.; El-Faham, A. Synthesis, Crystal Structure and DFT Studies of 1,3-Dimethyl-5-propionylpyrimidine-2,4,6(1H,3H,5H)-trione. Crystals 2017, 7, 31.

 Response 2

Comment from reviewer: I can’t figure out why details of Materials and Methods placed after Results and Discussion section. Maybe they should be swapped?

Reply from author: This is the preferred layout that the author would like the paper to be presented. Numerous other papers published in Crystals have followed this format. Therefore, the Editor will have to make the final decision if the current layout of the paper is acceptable or not.

Reviewer 3 Report

Crystals-410882

 Synthesis, Crystal Structure, DFT studies, Docking Studies and Fluorescent Properties of 2-(Adamantan- 1-yl)-2H-isoindole-1-carbonitrile

 Jacques Joubert

 This manuscript contains a description of the conditions used for the preparation of 2-(Adamantan-1-yl)-2H-isoindole-1-carbonitrile by chemical reaction under microwave stimulation. The crystal structure was defined by single crystal X-ray diffraction analysis. The molecular structure was confirmed by nuclear magnetic resonance, high-resolution mass spectroscopy and X-ray diffraction. The electronic structure was obtained by DFT calculations. The docking studies and fluorescence measurements were carried out by conventional methods. The experimental conditions used for the sample preparation and physical property measurements are accurately given that open a possibility for future testing of the results by other researchers. In my opinion, this study is interesting because the information is specified concerning preparation, structure and properties of the bioactive compound. The general level of this study is good and manuscript could be considered for publication after minor revision reasonable to increase the paper quality. My several proposed corrections and questions are listed below for author consideration.

 Page 1

 Compound 1 showed the highest fluorescence intensity in polar solvents with excitation and emission values of 336 nm and 380 nm, respectively.

 Compound 1 showed the highest fluorescence intensity in polar solvents with excitation and emission values of 336 and 380 nm, respectively.

 Page 1

 Radioligand binding techniques have been widely used to study biological proteins involved in the pathophysiology of neurodegenerative disorders such as Parkinson’s disease and Alzheimer’s disease [1-3].

 Radioligand binding techniques are widely used to study biological proteins involved in the pathophysiology of neurodegenerative disorders such as Parkinson disease and Alzheimer disease [1-3].

 Page 1

 Neuroprotective targets including the neuronal nitric oxide synthase (nNOS) enzyme [4], the N-methyl-D-aspartate (NMDA) receptor [3] and voltage gated calcium channels (VGCC) [5] have been widely studied using radioligands.

 Neuroprotective targets, including the neuronal nitric oxide synthase (nNOS) enzyme [4], the N-methyl-D-aspartate (NMDA) receptor [3] and voltage gated calcium channels (VGCC) [5], were widely studied using radioligands.

 Page 1

 Despite the usefulness and sensitivity of radioligand binding techniques the use of alternative approaches, such as fluorescent methods, to study receptor-ligand and enzyme-ligand binding interactions may provide information not readily accessible by conventional radiopharmacology.

 Despite the usefulness and sensitivity of radioligand binding techniques, the use of alternative approaches, such as fluorescent methods, to study receptor-ligand and enzyme-ligand binding interactions may provide information not readily accessible by conventional radiopharmacology.

 Page 2

 Compound 1 also has inherent fluorescent properties because of the inclusion of the cyanoisoindole fluorophore in its structure [11,13].

 Compound 1 also has inherent fluorescent properties because of the cyanoisoindole fluorophore in its structure [11,13].

 Page 2

 This study therefore set out to improve the synthesis of 1, study its crystal structure and geometry in both solid and free form state, analyse the frontier molecular orbitals and atomic net charges, perform docking experiments to elucidate the potential biological binding interactions and further explore its fluorescent properties.

 This study, therefore, set out to improve the synthesis of 1, study its crystal structure and geometry in both solid and free form state, analyse the frontier molecular orbitals and atomic net charges, perform docking experiments to elucidate the potential biological binding interactions and further explore its fluorescent properties.

 Page 2

 Both OPD and amantadine.HCl are commercially available.

 Supplier should be reported on for each starting reagent.

 Page 2

 The reaction proceeded efficiently providing the title compound in an excellent yield of 91% after only 10 minutes of reaction time.

 The reaction proceeded efficiently providing the title compound in an excellent yield of 91% after only 10 min of reaction time.

 Page 2

 This is a significant improvement in yield and reduction in reaction time compared to the conventional method previously described where a yield of 66% was obtained with a reaction time of 24 hours [12].

 This is a significant improvement in yield and reduction in reaction time compared to the conventional method previously described where a yield of 66% was obtained with a reaction time of 24 h [12].

 Page 2

 A crystalline material was grown from EtOH at room tempreture for 24 hours.

 A crystalline material was grown from EtOH at room temperature for 24 h.

 Page 3

 The X-ray refined data was used to select a crystal unit containing the starting geometries of compound 1.

 The X-ray refined data were used to select a crystal unit containing the starting geometries of compound 1.

 Page 3

 Table 1 contains the data comparing the experimental with the calculated structure of 1.

 Table 1 contains the data comparing the experimental and calculated structures of 1.

 Page 3

 For N2-C19 the calculated C≡N (nitrile) bond length is 1.16 Å which is longer than the experimental value of 1.14 Å and is exactly the same as the bond length (1.16 Å) described for nitriles in chemistry literature.

 For N2-C19, the calculated C≡N (nitrile) bond length is 1.16 Å which is longer than the experimental value of 1.14 Å and is exactly the same as the bond length (1.16 Å) described for nitriles in chemistry literature.

 Page 3

 The N1–C1, N1–C11 and N1–C14 bond lengths are calculated as 1.50, 1.40 and 1.36 Å at the B3LYP/6-311G level and experimental values are 1.49, 1.39 and 135 Å, respectively.

 The N1–C1, N1–C11 and N1–C14 bond lengths are calculated as 1.50, 1.40 and 1.36 Å at the B3LYP/6-311G level and the experimental values are 1.49, 1.39 and 135 Å, respectively.

 Page 3

 It is noted that the N1–C11 and N1–C14 bond lengths are much shorter than the normal C–N bond length (1.47 Å) while N1–C1 is closer to the normal length.

 It is noted that the N1–C11 and N1–C14 bond lengths are much shorter than the normal C–N bond length (1.47 Å), while N1–C1 is closer to the normal length.

 Page 4

 The bond angles calculated are in good agreement with the experimental data with a maximum deviation of 1.26° for C1–N1–C14, which is calculated as 125.31° and the experimental value is 126.57°.

 The calculated bond angles are in good agreement with the experimental data with a maximum deviation of 1.26° for C1–N1–C14, which is calculated as 125.31° and the experimental value is 126.57°.

 Page 4

 Overlay analysis was performed using YASARA version 18.11.21 (YASARA Biosciences GmbH) to globally compare the structure obtained with the theoretical calculation with that obtained from the X-ray diffraction (Figure 3).

 The overlay analysis was performed using YASARA version 18.11.21 (YASARA Biosciences GmbH) to globally compare the structure obtained with the theoretical calculation with that obtained from the X-ray diffraction (Figure 3).

 Page 5

 Figure 4 shows the distribution and energy levels of HOMO and LUMO orbitals computed at the B3LYP/6-311G level for the title compound 1.

 In Figure 4, the distribution and energy levels of HOMO and LUMO orbitals computed at the B3LYP/6-311G level for the title compound 1 are shown.

 Page 5

 As can be seen from Figure 4, both the LUMO and HOMO is mainly localised at the nitrile and isoindole moieties.

 As can be seen in Figure 4, both the LUMO and HOMO is mainly localised at the nitrile and isoindole moieties.

 Page 5

 However, even though the N1 atom showed a lower net atomic charge compared to N2 it will encounter some difficulty to form interactions with amino acids because of the structural hindrance imposed by the bulky adamantane group.

 However, even though the N1 atom showed a lower net atomic charge compared to N2, it will encounter some difficulty to form interactions with amino acids because of the structural hindrance imposed by the bulky adamantane group.

 Page 7

 It is important to note that 1 has only been evaluated on a crude NOS extract [12] containing the inducible-, endothelial- and neuronal NOS isoforms and further biological studies are necessary in order determine the NOS isoform selectivity profile of this compound.

 It is important to note that 1 has only been evaluated on a crude NOS extract [12] containing the inducible-, endothelial- and neuronal NOS isoforms and further biological studies are necessary in order to determine the NOS isoform selectivity profile of this compound.

 Page 7

 The excitation absorption maximum and the emission maximum were found to be 336 nm and 380 nm, respectively.

 The excitation absorption maximum and the emission maximum were found at 336 and 380 nm, respectively.

 Page 7

 When the electron-withdrawing nitrile group is present the charge separation of the molecule is enhanced due to an internal Stark effect [29,30] which, in turn, results in a red-shift in the fluorescence spectrum and increase in fluorecence intensity [31].

 When the electron-withdrawing nitrile group is present, the charge separation of the molecule is enhanced due to an internal Stark effect [29,30] which, in turn, results in a red-shift in the fluorescence spectrum and increase in fluorecence intensity [31].

 Page 7

 No noticeable shift in the emission value of 1 was observed in the different solvents.

 No noticeable shift in the emission maximum of 1 was observed in the different solvents.

 Page 8

 The reaction mixture was placed in the microwave reactor and irradiated at 150 W, 100 °C and 150 psi for 10 minutes.

 The reaction mixture was placed in the microwave reactor and irradiated at 150 W, 100 °C and 150 psi for 10 min.

 Page 8

 Upon cooling in an ice bath a white precipitate formed and was filtered and washed twice with cold methanol (10 ml) and twice with distilled water (10 ml) to yield the pure product (1) as a white amorphous solid in high yield (0.828 g, 2.997 mmol, 91%).

 Upon cooling in an ice bath, a white precipitate formed and was filtered and washed twice with cold methanol (10 ml) and twice with distilled water (10 ml) to yield the pure product (1) as a white amorphous solid in high yield (0.828 g, 2.997 mmol, 91%).

 Page 9

 The docking algorithm which was chosen for these experiments were based on induced fit docking to allow for flexible interactions of the test ligand with the protein.

 The docking algorithm, which was chosen for these experiments, was based on induced fit docking to allow for flexible interactions of the test ligand with the protein.

 Two questions are concerning crystal structure determination and data reported in the cif file:

 1) The Flack parameter is too low -2.5(10). It should be in the in the range of 0-1. Thus, it should be corrected or some comments are needed.

 2) There is large positive residual density equal to 0.69 eA-3. Can it be accounted or explained?

Author Response

Response 1

Grammatical errors as pointed out by the reviewer, where the author agrees with the reviewer, have been corrected throughout the manuscript. Listed below are the corrections that have not been made with the explanation as to why we these have been omitted.

Reviewer: Both OPD and amantadine.HCl are commercially available. Supplier should be reported on for each starting reagent.

Author response: The suppliers have been stated in the experimental section (see 3.1.1.).

Response 2

The two questions concerning the crystal structure determination and data reported in the cif file are addressed below.

Question 1

Comment from reviewer: The Flack parameter is too low -2.5(10). It should be in the in the range of 0-1. Thus, it should be corrected or some comments are needed.

Response from author: In this case the Flack parameter is meaningless since the molecule is achiral and crystallised in a chiral space group. The esd is also more than three times the parameter value so it is of no relevance. 

Question 2

Comment from reviewer: There is large positive residual density equal to 0.69 eA-3. Can it be accounted or explained?

Response from author: I am aware of the positive residual density and this is one of the reasons why the data collection was repeated a number of times. This is actually a pretty low electon density count and would normally be a hydrogen atom. Given it's location this is not likely and attempts were made to model the electron density as a partially occupied water molecule (or at least an oxygen atom), but this was also not correct. I believe this peak is a result of a poorly diffracting crystal - because this molecule is poorly organic it does not diffract to a very high resolution. In addition the data was collected more than once, however the CIF deposited was the best of the three.